# Integration of Rice Husk Ash as Supplementary Cementitious Material in the Production of Sustainable High-Strength Concrete

**DOI:** 10.3390/ma15228171

**Published:** 2022-11-17

**Authors:** Noor Md. Sadiqul Hasan, Md. Habibur Rahman Sobuz, Md. Munir Hayet Khan, Nusrat Jahan Mim, Md. Montaseer Meraz, Shuvo Dip Datta, Md. Jewel Rana, Ayan Saha, Abu Sayed Mohammad Akid, Md. Tanjid Mehedi, Moustafa Houda, Norsuzailina Mohamed Sutan

**Affiliations:** 1Department of Civil Engineering, College of Engineering and Technology, International University of Business Agriculture and Technology, Dhaka 1230, Bangladesh; 2Department of Building Engineering and Construction Management, Khulna University of Engineering and Technology, Khulna 9203, Bangladesh; 3Faculty of Engineering & Quantity Surveying, INTI International University (INTI-IU), Persiaran Perdana BBN, Putra Nilai, Nilai 71800, Negeri Sembilan, Malaysia; 4Graduate Research Assistant, Civil Engineering, Arkansas State University, Jonesboro, AR 72467, USA; 5College of Engineering and Technology, American University of the Middle East, Egaila 54200, Kuwait; 6Department of Civil Engineering, University Malaysia Sarawak, Kota Samarahan 94300, Sarawak, Malaysia

**Keywords:** rice husk ash, high strength concrete, sustainable concrete, mechanical properties

## Abstract

The incorporation of waste materials generated in many industries has been actively advocated for in the construction industry, since they have the capacity to lessen the pollution on dumpsites, mitigate environmental resource consumption, and establish a sustainable environment. This research has been conducted to determine the influence of different rice husk ash (RHA) concentrations on the fresh and mechanical properties of high-strength concrete. RHA was employed to partially replace the cement at 5%, 10%, 15%, and 20% by weight. Fresh properties, such as slump, compacting factor, density, and surface absorption, were determined. In contrast, its mechanical properties, such as compressive strength, splitting tensile strength and flexural strength, were assessed after 7, 28, and 60 days. In addition, the microstructural evaluation, initial surface absorption test, = environmental impact, and cost–benefit analysis were evaluated. The results show that the incorporation of RHA reduces the workability of fresh mixes, while enhancing their compressive, splitting, and flexural strength up to 7.16%, 7.03%, and 3.82%, respectively. Moreover, incorporating 10% of RHA provides the highest compressive strength, splitting tensile, and flexural strength, with an improved initial surface absorption and microstructural evaluation and greater eco-strength efficiencies. Finally, a relatively lower CO_2_-eq (equivalent to kg CO_2_) per MPa for RHA concrete indicates the significant positive impact due to the reduced Global Warming Potential (GWP). Thus, the current findings demonstrated that RHA can be used in the concrete industry as a possible revenue source for developing sustainable concretes with high performance.

## 1. Introduction

Concrete has been primarily recognized as the most frequently utilized construction material for several decades. Concrete production requires a lot of raw materials, which leads to a significant amount of energy consumption and pollution of nature [1]. Cement is one of the fundamental ingredients required in concrete construction, but its manufacturing leads to high CO_2_ emissions, which pollute the atmosphere [2]. The generation of one ton of cement is anticipated to emit approximately one ton of CO_2_, with 5–7% of worldwide carbon emissions resulting in environmental deterioration [3,4]. Furthermore, cement manufacturing consumes a great deal of energy, making it incompatible for long-term development [5]. To overcome such ecological and financial concerns, changes and enhancements to the current manufacturing technologies of concrete are necessary.

The concept has prompted concrete science and engineering researchers to look into and explore supplemental waste components that could be employed as alternatives for different elements in concrete manufacturing [6,7,8,9]. Waste disposal is another serious ecological challenge [3]. The use of waste products such as agricultural and industrial residues in concrete manufacturing lowers the ecological damage of inappropriate dumping, including the mounds of these wastes loaded in or discarded on fertile land. Since a variety of agricultural and industrial residues have characteristics that are acceptable for concrete manufacturing, a detailed examination of their possible use in making concrete offers significant potential to improve resource efficiency. Agricultural as well as other wastes such as palm oil fuel ash [10], sugarcane bagasse ash [11], oyster shell [12,13], sawdust [14,15], groundnut shell [16,17], coconut shell [18,19], tobacco waste [20,21], and so many more have been demonstrated to efficiently perform as a partial substitute inside the concrete mass, expanding the concrete properties satisfactorily.

Rice is a significant agricultural commodity in Asia [22,23]. Its production has increased significantly in the last decade, with worldwide paddy output expected to reach 760 million tons in 2017 [24]. Furthermore, as the global population is projected to continue expanding through 2050, rice production is likely to increase further [25]. Rice husk is a residue produced during the processing of rice that contributes roughly 20%–23% of the overall grain weights [22,23], implying that over 150 million tons are generated yearly. Unfortunately, around 83% of this amount is just thrown away as waste [25] that damages water, soil, and the overall environment [22,26,27]. Rice husk ash is a widespread agricultural waste that is a component leftover from the rice husk’s ignition process, which has received extensive attention from several studies [28,29]. The ash is frequently dumped in water bodies and landfills, endangering nature. Consequently, it constitutes a safety risk and significantly damages the environment. That is why various sectors are paying more attention to the adequate treatment and application of RHA.

Rice husk contains a high concentration of SiO_2_, making it ideal for usage as a pozzolan material [30]. Pozzolans are siliceous (sometimes siliceous and aluminous) substances with no or little cementing characteristics on their own. To generate cementitious compounds, they can interact with calcium hydroxide chemically at room temperature, in the company of moisture, in a finely dispersed form [31]. Rice husk can be burned at a temperature below 700 °C to obtain a decent-quality RHA, [32] which can be utilized as an additional cementing material or as a filler in paint and rubber [22,23]. This form of RHA contains a high silica concentration and has a minimal loss on ignition (LOI). RHA is a pozzolanic substance with a high reactivity that has already been effectively utilized in concrete as a substitution for cement [33,34,35] and silica fume [36,37], without sacrificing durability or strength properties.

The favorable benefits of RHA on concrete’s mechanical performance and longevity have been extensively studied [36,38,39,40]. Ganesan et al. [41] discovered that cement incorporating up to 30% RHA could be manufactured, without compromising the porosity or the strength. They also found that cement with 30% RHA showed approximately 35%, 28%, and 75% reduction in the concrete’s permeability, chloride diffusion, and chloride permeation, respectively. The interaction of RHA with Ca(OH)_2_ during the hydration phase of concrete led to the formation of additional C-S-H gel, which considerably increases the strength and durability characteristics. Ge et al. [42] considered the combined classification models that combine optimization techniques, such as particle swarm optimization, which provide a new level of understanding of the characteristics. Yu et al. [43] observed that the amorphous silica of RHA interacts with Ca(OH)_2_ for generating a specific type of C-S-H gel (Cal.5SiO_3_.5.×H_2_O) at a temperature of about 40 °C, with the contact of moisture. According to Bouzoubaâ and Fournier [44], substituting 7.5% to 12.5% RHA with ordinary Portland cement massively enhanced the concrete’s resistance to chloride-ion penetration. Similarly, mahmud et al. [45] calculated that by combining RHA and superplasticizer, high-strength concrete (HSC) of 80 MPa compressive strength could be formed at 14 days. RHA improved the compressive strength by approximately 40% over the age of 56 days, according to Nehdi et al. [46], and was more preferable than silica fume. It was stated that about 30% RHA with cement not only improved the strength properties and corrosion resistance but also decreased the permeability and chloride penetration [47]. Giaccio et al. [48] found that the RHA particles lessen the porosity of concrete, particularly at the weakest zones of the material—the interfaces between cement paste and aggregates. In addition, they ensure practical usage of agricultural residues and minimize the amount of energy consumed in cement manufacture.

The investigation was directed to reinforce a link between the two research topics, namely the incorporation of waste products in concrete and the demonstration of their relevance. Furthermore, RHA is not frequently employed throughout the Malaysian building industry. The fundamental goal of the study is to evaluate the behavior and performance of high-strength concrete incorporating RHA at various proportions ranging from 0% to 20%, as well as to promote the utilization of waste materials for a sustainable future. Various fresh properties including slump, compacting factor, density, compressive strength, splitting tensile strength, and flexural strength were all evaluated, and the strength properties were compared with conventional HSC. Furthermore, an environmental impact assessment of the produced HSC was investigated.

### Significance of This Study

With growing environmental consciousness at all levels of society, the pollution and health hazards specifically associated with the concrete, cement, and clay brick industries are coming under intense scrutiny from environmentalists and governments. A significant amount of RHA is generated in Malaysia, which imposes tremendous environmental challenge. Even worse, the high population growth and shortage of waste-disposal sites in Malaysia pose a challenge to deal with this kind of toxic waste. They pose environmental problems such as air pollution and the leaching of hazardous and toxic chemicals (arsenic, beryllium, boron, cadmium, chromium, chromium (VI), cobalt, lead, manganese, mercury, molybdenum, selenium, strontium, thallium, and vanadium, along with dioxins, polycyclic aromatic hydrocarbon compounds, etc.) when dumped in landfills, quarries, rivers, and oceans. Consequently, air and water pollution have been inextricably linked to environmental problems and climate change. This study was conducted to maximize the application of RHA, which may help to reduce the environmental challenge, and to find a cost-effective sustainable solution for producing high-strength concrete.

## 2. Materials and Methods

Ordinary Portland cement (OPC) was collected from local vendors and considered the prime binder material for the concrete mixes that corresponded to the standard BS EN 196-1:2016 [49]. The cement was Type I OPC, fulfilling the requirements of every aspect. Along with OPC, rice husk ash (RHA) is employed as a subsidiary cementitious agent in quantities ranging from 0% to 20%. The chemical composition of the binders (ordinary Portland cement and RHA) utilized in the investigation is shown in Table 1. Chemical composition was done by X-ray fluorescence (XRF) to find the percentages of chemical constituents.

This study’s experimental procedure includes the evaluation of fresh properties, mechanical properties, microstructural properties, environmental impact, and cost–benefit assessment. So, the sequential works are represented in this study, as shown in Figure 1.

Rice husk was gathered from a mill located in Kuala Selangor, Malaysia, and incinerated inside a ferrocement furnace at the University of Malaya to generate RHA. After about an hour of keeping the flame beneath the furnace, the husk began to smolder itself for around 24 h. Then, the ash was kept for settling within the furnace for the next 24 h, before being extracted and pulverized with a Los Angeles machine. The specimen of rice husk ash utilized in this research is illustrated in Figure 2.

Physical properties also can be understood by the particle-size distribution curve. The distribution curve of RHA is shown in Figure 3, considering the unit as the micro-meter. Values for the particle distribution curve re taken from the hydrometer test analysis. The result ranges from 1.0 µm to around 60 µm.

This investigation used crushed aggregates from the quarry as coarse aggregate (CA), collected from local suppliers with a nominal size of 20 mm, complying with BS 812-103.1:1985 [50]. The fineness modulus was found to be 5.94. The physical properties of coarse aggregates are highlighted in Table 2. In contrast, the fine aggregate was obtained from the local market and air-dried to achieve the saturated surface dry status to maintain the water–cement ratio. The characteristics of fine aggregate are also represented in the below table. These properties were determined per ASTM C29, ASTM C127, ASTM C128, and ASTM C136 [51,52,53,54].

Figure 4 depicts the gradation curve of coarse and fine aggregate derived from the sieve analysis, complying with ASTM C33 [55]. The graph also represents the comparison of used materials with the standard ASTM lower and upper limit for coarse and fine aggregates. As a high-range water-reducing admixture, R1000 superplasticizer was utilized that is dark brown, chloride-free sulfonated naphthalene formaldehyde, which can dissolve in water and is 40% solid content with 1.2 specific gravity. Potable water was used to make all of the concrete combinations.

## 3. Experimental Program

### 3.1. Mix Proportioning for the Specimens

This experimental investigation was done for five concrete mixes prepared by altering the water–binder ratio and superplasticizer concentrations. The water–binder ratio and the superplasticizer remained constant at 0.35 for all concrete mixes, which are about 0.5% of the cement on a weight basis. Five mixtures of the concrete were generated in the laboratory to conduct this research. The initial mix was termed the reference (control) mix, as it was prepared with a conventional mix design and without any RHA, and was designated as RHA0 for this investigation. The remaining four mixes had the cement substituted by 5%, 10%, 15%, and 20% RHA and were termed RHA5, RHA10, RHA15, and RHA20, respectively. The concrete mixtures with various rice husk ash percentages are demonstrated in Table 3.

### 3.2. Preparing Specimens and Curing

All concrete mixes were designed with a water–binder ratio of 0.275. A 0.5% weight-based addition of superplasticizer was made to the cement. According to ASTM C685M-17 [56], the concrete specimens were prepared by mixing all materials in the correct proportions. Free-fall mixing machines with fixed drums were used in this case. The specimens were prepared and cured in accordance with ASTM C192M-19 [57] mixing procedures.

The compressive strength tests were carried out on a cube specimen of 100 mm × 100 mm × 100 mm, whereas the splitting tensile strength tests were performed on a cylindrical specimen, with a diameter of 100 mm and a height of 200 mm. The flexural strength was determined utilizing 100 mm × 100 mm × 500 mm prisms prepared in the lab. The concrete in the molds was temped in three stages, while casting to achieve optimal compaction and void elimination. Before demolding, all samples were allowed to stand for 24 h to ensure they were sufficiently hardened. The samples were removed from the molds and labeled for accurate recordkeeping after 24 h. Then, all the specimens emerged in curing tanks completely at 27 ± 2 °C ambient temperatures for 7, 28, and 56 days.

### 3.3. Testing Methods

The slump test was performed for the reference and other mixtures employing the ASTM C-143 [58] standard, to assess the effect of RHA on concrete workability. In addition, the compacting factor test was performed in compliance with BS EN 12350-4:2009 [59], to further comprehend the workability of concrete mixtures. In this investigation, the density of the concrete mix was evaluated per BS EN 12390-7:2009 [60]. In contrast, the initial surface absorption test was utilized for determining the surface absorption of the concrete integrating various levels of RHA, according to BS 1881-208:199 [61]. The compressive strength was evaluated at 7, 28, and 60 days by employing a 100 mm × 100 mm × 100 mm cube, in conformance with the BS EN 12390-3:2009 standard [60]. The splitting tensile strength test was carried out on 100 mm × 200 mm cylinders, in compliance with BS 1881-117:1983 [62], whereas BS EN 12390-5:2019 [63] was adopted for assessing the flexural strength of concrete, via a simple prism of 100 mm × 100 mm × 500 mm in size, employing third-point flexural loading at 7, 28, and 56 days. Figure 5 shows all testing setups considering the fresh tests; hardened tests in the laboratory maintain the mentioned standard configurations.

## 4. Results and Discussion

### 4.1. Fresh Properties

#### 4.1.1. Slump and Compacting Factors

Figure 6a provides a graphical illustration of the compacting factors and slump values of all concrete mixes. According to the results, the slump value tended to decline for the rising concentration of RHA in the concrete mix. The slump of the reference mix with 0% RHA was determined to be around 56 mm, whereas the slump of RHA5 was reported to be 47 mm, which is 9 mm smaller than the reference mix. Furthermore, for 10%, 15%, and 20% RHA-incorporating concrete, the slump values were around 39.5 mm to 34 mm, 30.5 mm to 16.5 mm, and 22 mm to and 25.5 mm, respectively, lower than the RHA0 concrete mix.

The compacting factor test of concrete mixture, equivalent to the slump test, exhibits a downward tendency when the proportion of RHA increases, as shown in Figure 6a. The value of the compacting factor for the reference mix RHA0 was about 0.943, whereas for the RHA5, RHA10, RHA15, and RHA20 mixes, it was about 0.911, 0.895, 0.88, and 0.865, respectfully. The equivalent slump values for the RHA-incorporated mixes were altered from 56 mm to 30.5 mm, based on the compacting factor of the concrete, as previously described. Furthermore, compared to the reference mix, the mixes incorporating 5%, 10%, 15%, and 20% RHA experienced a decline in compaction factor of 3.39%, 5.09%, 6.68% and 8.27%, respectively.

#### 4.1.2. Fresh Density

The relation between the density and slump value of the concrete mixtures is depicted in Figure 6b. It can be seen that when the quantity of RHA was raised, the concrete density declined progressively. The density of these concrete mixtures was evaluated in the spectrum of roughly 2638 to 2467 kg/m^3^, with associated slump ranging from 56 to 30.5 mm. The density of the control sample was about 2638.125 kg/m^3^, whereas the 5%, 10%, 15%, and 20% RHA concrete mixes had densities of 2600.63 kg/m^3^, 2565 kg/m^3^, 2509.38 kg/m^3^, and 2466.875 kg/m^3^, respectively, exhibiting 1.42%, 2.77%, 4.88%, and 6.49% reduction compared to the RHA0 mix. These findings are consistent with the investigation of Dong et al. [64] and Ephraim et al. [65]. They found that when the proportion of rice husk ash increases, the density declines. Habeeb and Mahmud [66] investigated the mechanical behavior of rice husk ash concrete and discovered a gradual decline in density, ranging from 2347 to 2253 kg/m^3^. The reason behind the reduction is the reduced specific gravity of RHA. It is substantially lower than cement, leading to a decrease in mass per unit volume of the concrete mixes.

### 4.2. Mechanical Properties

#### 4.2.1. Compressive Strength

Table 4 presents the compressive strength for all concrete mixes with various proportions of rice husk ash with reference to mean compressive strength, coefficient of variation (COV), standard deviation, standard error, and 95% confidence interval within 7, 28, and 56 days. It is observed from the table that the compressive strength varied from 13.23 to 22.92 MPa, 19.25 to 27.98 MPa, and 22.92 to 36.93 MPa on the 7th, 28th, and 56th days, respectively. Concrete’s standard deviation varied from 0.15 to 1.33, along with a coefficient of variation from 0.32% to 2.08% and a standard error of 0.09 to 0.77. At seven days, RHA20 possessed the minimum compressive strength of 41.476 MPa, including a 95% confidence interval of 40.87 MPa to 42.08 MPa, whereas RHA10 showed the maximum compressive strength of 47.04 MPa, including a 95% confidence interval of 46.89 MPa to 47.19 MPa. Similar results were found at 28 and 56 days, when the RHA20 had the minimum compressive strength of 61.50 MPa and 63.77 MPa, respectively, and RHA10 showed the best results of 68.92 MPa and 71.85 MPa, respectively.

Figure 7 illustrates the compressive strength of the test specimens incorporating a different proportion of RHA. Concrete’s compressive strength increased as the proportion of RHA climbed up to 10% RHA, and after that it dropped with the increment of RHA. Among all concrete mixtures, the RHA10 mix obtained the highest compressive strength, which was 47.04 MPa, 68.92 MPa, and 71.85 MPa for 7, 28, and 56 days, respectively.

Several researchers have suggested that the increase in compressive strength could be due to various factors, such as RHA’s microfilling ability and pozzolanic activity. Moreover, RHA is highly reactive; it reacts with calcium hydroxide (a byproduct of cement hydration) and produces additional C-S-H. The additional C-S-H reduces the porosity of concrete by filling the capillary pores, thus improving the concrete microstructure in the bulk paste matrix and transition zone, leading to increased compressive strength. 

A decrease in compressive strength was observed with mixes containing more than 10% of RHA. Various researchers also reported such a trend in strength. Kannan et al. [67] reported that the compressive strength of SCC blended with 15% RHA was higher than that of normal SCC. As per the findings of Chopra et al. [68], the compressive strength increases with increases in the percentage of RHA up to 15% replacement; after that, the strength starts to decrease. The reason behind this could be that the amount of silica available in the hydrated blended cement matrix was probably too high, and the amount of the produced C-H was most likely insufficient to react with all the available silica. As a result, some amount of silica was left without any chemical reaction [68]. Another reason could be that due to the replacement of cement with RHA, water demand is increased, adversely affecting the concrete’s strength [69,70].

Figure 8 illustrates the change (%) in strength properties for a different proportion of the RHA incorporation mix than the reference mix. According to findings, RHA5 and RHA10 increase the concrete’s compressive strength, whereas succeeding RHA mixtures possess lower strength than the reference mixture RHA0. The RHA10 concrete mix maintained the most significant improvement of 2.25%, 6.01%, and 7.16% at 7, 28, and 56 days, respectively, whereas the RHA20 concrete mix showed approximately 9.83%, 5.39%, and 4.88% of compressive strength reduction at 7, 28 and 56 days, respectively, in comparison with the reference mix RHA0. Similar findings was obtained by Alex et al. [26], who claimed that the 10% RHA inclusion leads to a significant compressive strength rising with respect to the reference mix.

#### 4.2.2. Splitting Tensile Strength

Table 5 summarizes the splitting tensile strength for concrete mixes with various proportions of rice husk ash, with reference to mean tensile strength, coefficient of variation (COV), standard deviation, standard error, and 95% confidence interval within 7, 28, and 56 days. The result demonstrates that the outcomes of splitting tensile strength fluctuated from 4.19 MPa to 4.71 MPa, 6.19 MPa to 6.79 MPa, and 6.39 MPa to 7.16 MPa on the 7th, 28th, and 56th day, respectively. Concrete’s standard deviation varied from 0.01 to 0.56, along with a coefficient of variation differing from 0.15% to 1.33% and a standard error of 0.01 to 0.03. At 7 days, RHA20 showed the minimum result, of about 4.19 MPa including a 95% confidence interval of 4.15 to 4.23 MPa, whereas RHA10 had the greatest tensile strength of 4.71 MPa, including a 95% confidence interval of 4.69 to 4.74 MPa. Similar results were found at 28 and 56 days, when RHA20 possessed the minimum results of 6.19 MPa and 6.38 MPa, respectively, while RHA10 depicted the maximum results of 6.79 MPa and 7.16 MPa, respectively.

Figure 9 shows a graphical representation of splitting tensile strength for the concrete mixtures with increasing RHA concentrations at 7, 28, and 56 days. It is observed that when the proportion of RHA in the mix proportion grew up to 10%, the splitting tensile strength climbed, and, afterwards, it dropped at an acceptable rate. At 7, 28, and 56 days, the reference mix RHA0 produced splitting tensile strengths of 4.59 MPa, 6.50 MPa, and 6.68 MPa, respectively. The RHA10 mix had the greatest splitting tensile strength of 4.71, 6.79, and 7.16 MPa at 7, 28, and 56 days, respectively.

Compared with the reference concrete, RHA20 possessed a splitting tensile strength of 4.19, 6.19, and 6.37 MPa at 7, 28, and 56 days, respectively, which is comparable to the findings of earlier researchers [40,41,66,71]. They described that the lesser RHA percentages resulted in a greater splitting tensile strength than the reference concrete, whereas greater RHA levels led to reduced strength. The rise in the pozzolanic reaction among the RHA and the cement constituents in the samples provides more tensile strength for concrete. Due to the tiny particles of the RHA, the porous areas inside the concrete turns into a denser matrix. Therefore, the bonding between the aggregates and the cement paste becomes stronger, providing the additional strength required to withstand the tension.

Figure 10 demonstrates the change (%) in splitting tensile strength for the different proportions of the RHA-incorporated mix, compared to the reference concrete RHA0. The outcomes of the tests revealed that RHA5 and RHA10 enhanced the concrete’s splitting tensile strength; nevertheless, consecutive RHA mixtures displayed lower strength than RHA0. Compared with the reference concrete mix, the RHA10 mix had the maximum splitting tensile strength enhancement of 2.45%, 4.52%, and 7.03% at 7, 28, and 56 days, respectively, while the RHA20 mix exhibited approximately 8.92%, 4.81%, and 4.58% of strength reduction at 7, 28, and 56 days, respectively. Saraswathy and Song [47] revealed that the splitting tensile strength improvement continued until 25%, while the maximum percentage enhancement recorded in the concrete containing 15% RHA was about 9.58% at 28 days.

#### 4.2.3. Flexural Strength

The flexural strength for RHA-incorporated concrete mixes regarding mean flexural strength, coefficient of variation (COV), standard deviation, standard error, and 95% confidence interval within 7, 28, and 56 days are depicted in Table 6. The research revealed that when the proportion of the RHA grew to 10%, the concrete’s flexural strength increased and eventually decreased with the RHA percentage increment. The flexural strength changed from 4.87 MPa to 5.33 MPa, 7.69 MPa to 8.15 MPa, and 8.97 MPa to 9.45 MPa on the 7th, 28th, and 56th day, respectively. The standard deviation of the examined sample spanned from 0.01 to 0.31, along with a COV varying from 0.26 % to 5.57% and a standard error of 0.01 to 0.18. At 7 days, the RHA10 showed the optimal flexural strength of around 5.33 MPa, including 95% confidence interval of 5.28 MPa to 5.35 MPa. In contrast, the RHA20 had a minimal flexural strength of 4.87 MPa, including a 95% confidence interval of 4.78 MPa to 4.94 MPa. Analogous outcomes were detected at 28, and 56 days, where the RHA10 possessed the strength of 8.15 MPa, and 9.45 MPa at 7 days, respectively, while the RHA20 had the results of 7.69 MPa and 8.97 MPa, respectively at seven days.

The obtained flexural strength for all the concrete mixes with the varying RHA concentrations at 7, 28, and 56 days is illustrated in Figure 11. The flexure strength was observed to rise when the RHA’s proportion climbed until 10% and eventually declined with the increment of the RHA percentage. Flexural strength exhibits a similar pattern to compressive and splitting tensile strengths. In the reference mix, RHA0 produced the flexural strength of 5.18 MPa, 7.94 MPa, and 9.10 MPa at 7, 28, and 56 days, respectively. Comparing among all other samples, the RHA10 mix was found to exhibit the maximum flexural strength of 5.33, 8.15 and 9.45 MPa at 7, 28 and 56 days, respectfully. Conversely, the RHA20 mix had the minimum flexural strength of 4.87, 7.69, and 8.97 MPa at 7,28, and 56 days. This observation is harmonic with other studies [40,66,72].

Figure 12 shows the % improvement in the flexural strength of the mixtures with respect to the reference mix. The RHA10 possessed the maximum flexural strength and surpassed the reference mix by 2.72%, 2.71% and 3.82% at 7, 28 and 56 days, respectively. Contrarily, the RHA20 concrete mix drifted from the reference mix by 6.09%, 3.02%, and 1.49% at 7, 28, and 56 days, respectively, in terms of flexural strength. The elevated pozzolanic interaction between the binder cement and constituents of RHA in samples and the packing efficiency of the RHA fine particulates might be the key reasons. Furthermore, this study was conducted using a low water/cement ratio of 0.35, which may result in increased flexural strength.

### 4.3. Initial Surface Absorption of Concrete

Figure 13 and Figure 14 depicts 56- and 91-day initial surface absorption test of the concrete integrating various levels of RHA. It can be seen that when the proportion of RHA substitution increases, the permeability declines. Furthermore, the permeability of RHA decreased as the measurement frequency increased for 10, 30, 60, and 120 min, respectively, owing to the saturation of the gaps throughout the testing period. At 56 days, after 10 min from the commencement of the experiment, the reference RHA0 specimen acquired 0.24 mL/m^2^ s, which then reduced to 0.19 mL/m^2^ s after 30 min of measuring. Surface absorption for 10%-RHA replacement fell to 0.16 mL/m^2^ s at the 10 min mark, then steadily climbed to 0.26 and 0.29 mL/m^2^ s for concrete with 15% and 20% RHA (Figure 13). On the other hand, the findings of the 91-day surface absorption showed the same decrease as the 56-day absorption. After 10 min of the test, the reference RHA0 samples obtained 0.21 mL/m^2^ s, which decreased to 0.16 mL/m2.s after 30 min of measurement. In addition, at the 10 min mark, surface absorption for sample RHA10 decreased to 0.13 mL/m^2^, while for samples RHA15 and RHA20, it rose to 0.24 mL/m^2^ and 0.26 mL/m^2^, respectively (Figure 14). Dhir et al. [73] observed identical results in the initial surface absorption of concrete (ISAT) values of concrete samples. This could be because of decreased pores inside the structure, since the pozzolanic material (RHA) fills throughout the blank spaces, lowering the concrete’s permeability. This outcome is consistent with the fact that adding RHA reduced the absorption feature, and this decline improved as the RHA proportion rose [74].

### 4.4. Characterization of Microstruicture

Microstructural analyses of concrete specimens were done by SEM (scanning electron microscope). The original microstructure and morphology of the hydrate mixes were visible on the crushed sample surfaces. Gold-coated crushed samples were placed on the brass stubs using carbon adhesives. Figure 15 shows the micrographs of (a) RHA0 (control), (b) RHA10, and (c) RHA20 at a 10 µm scale. The first one shows the irregular edges and platelets of cementitious particles, due to the incomplete activation of the superplasticizer. Clearly, some differences are identified between the graphs after using the RHA at various concentrations. Figure 15b describes the porous conditions using RHA. It shows dense microstructures with fewer cracks at an ambient temperature. The 10% replacement of RHA improves the microstructure with spherical shapes of concrete mix by filling pores in the next graph. Another one, incorporated with 20% RHA, develops homogeneous and dense microstructures with very few pores, resulting in good characteristics. Agglomerated states can be seen here, but the flakiness helps the concrete to consider the reduction in sorptivity value. The amount of C-A-S-H gel in the matrix rises with the addition of RHA. RHA produced dense particle packing with refined pore sizes and, as a result, dense microstructure, since it is good in quality. The micro-filler effect of RHA, which stops the pores and lowers mix permeability, causes this decrease in sorptivity value. The other literature considering RHA replacement with microstructural analysis gives satisfying results that justify this current study [75]. The siloxy bridges (Si-O-Si) chains formed by the SiO_2_ content of RHA decrease the gaps of the C-S-H gel and firmly link the particles, creating the dense microstructure visible in the micrograph. These micrographs support the findings from the preceding section’s analysis of the mechanical strength of the ideal combination.

### 4.5. Relationships between the Strengths

The interrelationship between the compressive and flexural strength of the concrete incorporating various proportions of RHA for 7, 28, and 56 days is illustrated in Figure 16. The correlation can be used for the flexural strength’s prediction of the concrete mixtures incorporating RHA. Various standards prescribe different alternative formulae to forecast flexural strength by relying on the compressive strength. ACI 363R, ACI 318, CEB-FIP, and AS 3600 [76,77,78,79] standards were employed to determine the flexural strength of concrete, with respect to the different compressive strength formulas.
(1)$$f_{fr}=0.60\;\sqrt{f_{c}}\;\;\left(AS\;3600\right)$$
(2)$$f_{fr}=0.62\sqrt{f_{c}}\;\;\left(ACI\;318\right)$$
(3)$$f_{fr}=0.94\sqrt{f_{c}}\;\;\left(ACI\;363R\right)$$
(4)$$f_{fr}=0.46\;{\left(f_{c}\right)}^{2/3}\;\left(CEB-FIP\right)$$
where *f_fr_* = flexural strength (MPa), *f_c_* = compressive strength (MPa).

According to the relationship graph, the ACI 363R formula and the CEB-FIP formula are probably the best predictors of flexural strength among the other codes of standardized formulas for RHA-based concrete. Contrarily, AS 3600 showed the most significant deviation from the experimentally anticipated flexural strengths and significantly undervalued the flexural strength. The 0.96 coefficients of determination (R^2^) demonstrate a good linear correlation between the compressive and flexural strengths for various durations at 7, 28, and 56 days. The preferred formula for representing the relation between the compressive and flexural strength is as follows:(5)$$f_{fr}=0.6461\;f_{c}+4.803$$

Likewise, Figure 17 highlights the correlation between the splitting tensile strength and compressive strength of the concrete mixes incorporating RHA. The forecast can be used in the finite element analysis to construct the RHA-based structural elements. This could improve materials and mix proportions without the need for expensive costs, long processing times, or bulky equipment. To forecast the splitting tensile strength depending on the compressive strength, ACI 363R, ACI 318, CEB-FIP, and AS 3600 [76,77,78,79] suggest Formulas (6)–(9), respectively.
(6)$$f_{st}=0.4\;\sqrt{f_{c}}\;\;\left(AS\;3600\right)$$
(7)$$f_{st}=0.56\sqrt{f_{c}}\;\;\left(ACI\;318\right)$$
(8)$$f_{st}=0.59\sqrt{f_{c}}\;\left(ACI\;363R\right)$$
(9)$$f_{st}=0.3\;{\left(f_{c}\right)}^{2/3}\;\left(CEB-FIP\right)$$
where *f**_st_* = splitting tensile strength (MPa) and *f’*_c_ = compressive strength (MPa).

It can be seen that amongst all the standards, the estimated values of ACI 363R showed the nearest value compared to the experimental ones, whereas AS 3600 showed the most prominent variation within the experimental results. A strong correlation was determined between compressive and splitting tensile strengths, with the coefficient of determination R^2^ = 0.99, indicating that 99% of the average tensile strength can be clarified through that linear correlation, while the residual 1% remains unexplained. In this investigation, the following formula can be advised for expressing the relationship between the splitting tensile and compressive strengths of RHA-incorporated concrete.
(10)$$f_{st}=0.0976\;f_{c}+0.145$$

## 5. Environmental Impact and Cost–Benefit Assessment

Cement production accounts for around 10% of worldwide CO_2_ emissions [80,81,82]. Due to the activities involved in preparing raw materials, concrete, the world’s most widely used building material, has a substantial carbon footprint. Since OPC is the primary binding material of concrete, it is responsible for between 75% and 90% of the total CO_2_ emissions [82] produced by concrete. The awareness of Global Warming Potential (kgCO_2_-eq/kg material) has enabled the development of alternative binders to reduce concrete’s reliance on OPC as its primary binder.

Although using SCMs will undoubtedly lower the overall embodied CO_2_ emissions of concrete, the embodied CO_2_ emissions of different waste materials will vary. Regarding the present investigation, RHA was partially utilized as a binder to replace OPC. To highlight the impact of RHA on the overall embedded CO_2_ emissions of HSC, this study estimates the total CO_2_ emissions, by considering the equivalent CO_2_ releases for each material. These numbers were extracted from relevant research studies and are presented in Table 7.

The embodied GWP for each mixture was calculated by multiplying the weight of each ingredient required to manufacture one cubic meter of concrete by the embodied CO_2_ of each material and then adding the results. As seen in Figure 18, the OPC has the highest CO_2_ content compared to other materials. OPC is responsible for roughly 89.10% of the total CO_2_ emissions for the control mix (RHA0). This was significantly reduced by substituting 20% OPC with RHA (RHA20). For the RHA20 mix, the CO_2_ contribution of OPC input decreased from 492 to 393.6 kg-CO_2_/m^3^. In addition, RHA5, RHA10, RHA15, and RHA20 reduced 4.31%, 8.63%, 12.95%, and 17.27% of the CO_2_ emissions, respectively.

The eco-strength efficiency of concrete is another metric that may be applied to the environmental impact assessment for evaluation. Alnahhal et al. [88] used the term eco-strength efficiency, while Damineli et al. [89] referred to it as the CO_2_ intensity, which can be defined as the quantity of CO_2_ emissions produced per unit of performance. It was determined using Formula (11)
(11)$$C_{i}=\frac{CO_{2}}{C_{s}}$$
where *C_i_* represents the eco-strength efficiency, which is the intensity of CO_2__,_ CO_2_ represents the embodied carbon dioxide emissions released by the concrete mixes as determined using Table 4, and *C_s_* represents the compressive strength.

The eco-strength efficiency of the various mixes is compared with their compressive strength to facilitate an organized and consistent comparison of the mixes. Figure 19 depicts the results of the study. The compressive strength values for 7 days, 28 days, and 60 days are represented by the bar chart from bottom to top, while the CO_2_ intensity is represented by the line along the secondary axis. The CO_2_ intensity allows for the evaluation of both the performance and contribution of concrete mixes to Global Warming Potential per their unit strength, which makes it a good predictor of the impact of concrete use (Damineli et al.) [89]. In most cases, the concentration of CO_2_ rises when there is a greater amount of Portland cement in the concrete mixture. Substituting cement with RHA, CO_2_ intensity can be reduced for a given strength, particularly at later ages. For instance, lowering the percentage of Portland cement from 100% (RHA0) to 20% (RHA20) resulted in a decrease in the CO_2_ intensity from 11.89 MPa/kgCO_2_-eq.m^−3^ to 8.1 MPa/kgCO_2_-eq.m^−3^.

Before adopting any newer or mixed concrete, the construction industry considers the environmental impact assessment and the cost–benefit analysis as two of the most critical factors. The cost–benefit analysis of concrete, which was the subject of this study, was computed for each mix, based on the costs of the individual materials in the area. Table 8 displays, in USD, the cost of one kilogram of each commodity when converted from BDT (1 USD = 93 BDT). The availability of various materials in a given region significantly impacts their respective pricing. Due to this, the price of RHA, a pozzolanic material, is considerably less than the cost of OPC. Figure 20 depicts the cost of each concrete mixture per 1 cubic meter. It can be seen that the RHA0 mix has a price of USD 164.28 per m^3^ of concrete. The OPC and the superplasticizer are the primary factors that determine the pricing. The RHA20 mix has the lowest overall cost, costing USD 149.02 per m^3^ of concrete produced. Compared to the control mix, this is approximately 9.28% less expensive.

As observed in Figure 21, the overall cost of each mix varies; however, a comprehensive cost–benefit analysis cannot be considered complete without assessing the cost to produce 1 MPa of strength for each mix. Therefore, the cost to produce a 1-MPa compressive strength at 28 days is calculated and shown in Figure 20. The lowest cost of producing 1 MPa was exhibited by the RHA10 mix, at 2.27 USD/MPa, with a reduction of 10.04% compared to the control mix. Moreover, the RHA5, RHA15, and RHA20 mixes depicted the cost index of 2.369, 2.390, and 2.423 USD/MPa, respectively.

## 6. Conclusions

The awareness of waste management and environmental issues has led to considerable breakthroughs in using waste materials such as rice husk ash. The fundamental purpose is to assess the fresh and mechanical properties of concrete, including RHA as a cement substitute, to determine their feasibility for construction and attempt to preserve natural resources, recycle trash, and make long-lasting concrete. Based on the experimental outcomes, the following significant observations and conclusions can be reached:The concrete mixture containing RHA exhibited poor overall workability due to a low water-to-cement ratio, a constant concentration of SP, and the cellular structure of RHA, which absorbed moisture.As the concentration of RHA was increased, the density of the hardened concrete mixtures decreased; similarly, the ISAT results indicated that the permeability of the concrete decreased as the concentration of RHA increased.The optimum compressive strength was achieved with RHA10, which was the 10% RHA supplementation by the weight of the cement. The reason could be due to the pozzolanic activity and pore refinement of the RHA particles.The addition of 10% RHA to concrete enhanced the splitting tensile strength compared to RHA0 (reference concrete), and the percentage of the splitting tensile strength enhancement ranged from approximately 2.0% to 7.5%. However, a decrease in the tensile strength was observed after 10% RHA incorporation.The flexural strength results were identical to the compressive and tensile strength tests. According to these determinations, concrete incorporating 10% RHA possessed maximum results of 5.3250, 8.1500, and 9.4517 MPa at 7, 28, and 56 days, respectively.The relationship between splitting tensile strength and flexural strength showed a good linear connection with the enhancement of compressive strength. The estimated values of ACI 363R and CEB-FIP were the closest among all standards for both flexural and splitting tensile strength, whereas AS 3600 possessed the highest fluctuation.Microstructural investigation of the specimens presented a mostly irregular and agglomerated state of the concrete. Chemical reactions and micrographs of the RHA-incorporated concrete depicted the microfiller effect. Based on the flakiness of the dense microstructure combination, RHA may achieve the strength in an incremental style up to 10% utilization, demonstrating the optimal consideration.RHA20 depicted the lowest GWP of 452.45 kgCO_2_-eq/kg material, with a 17.27% reduction than the control mix, whereas the RHA10 mix represented an 8.63% GWP reduction, while achieving 3.28% better compressive strength than the control mix, demonstrating a significant eco-strength efficiency.In terms of cost–benefit analysis, the RHA20 mix had the lowest overall cost, projected to be approximately 9.28% less expensive than the control mix. Although the RHA10 mix showed little cost reduction (only 4.64% than the control mix), it depicted the lowest cost index value of 2.27 USD/MPa, which is almost 11% cheaper than the RHA0 mix.Among all the mix proportions, the 10%-RHA-incorporated concrete produced overall satisfactory performance for its fresh and mechanical characteristics as well as in its environmental impact analysis and cost–benefit analysis. It is feasible to generate high-strength concrete (grade 60) by employing RHA as an additional cementing element, which leads to waste-product mitigation and a sustainable alternative for the construction field.

## Figures and Tables

**Figure 1 materials-15-08171-f001:**
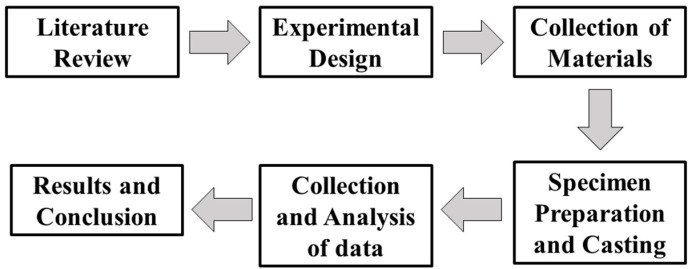
Schematic flowchart.

**Figure 2 materials-15-08171-f002:**
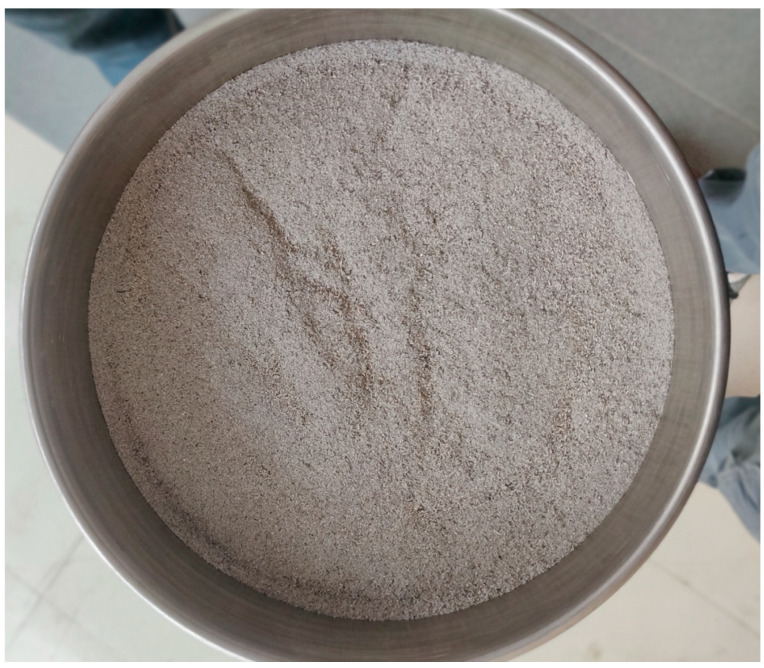
Sample of rice husk ash.

**Figure 3 materials-15-08171-f003:**
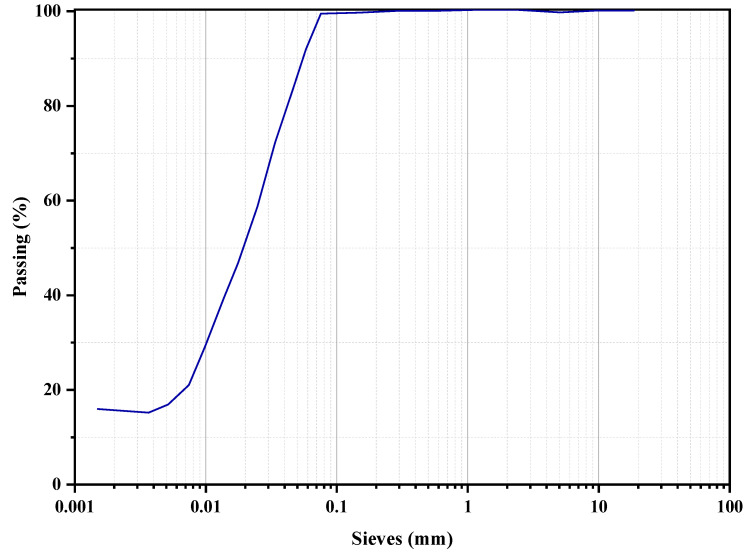
Particle size distribution of RHA.

**Figure 4 materials-15-08171-f004:**
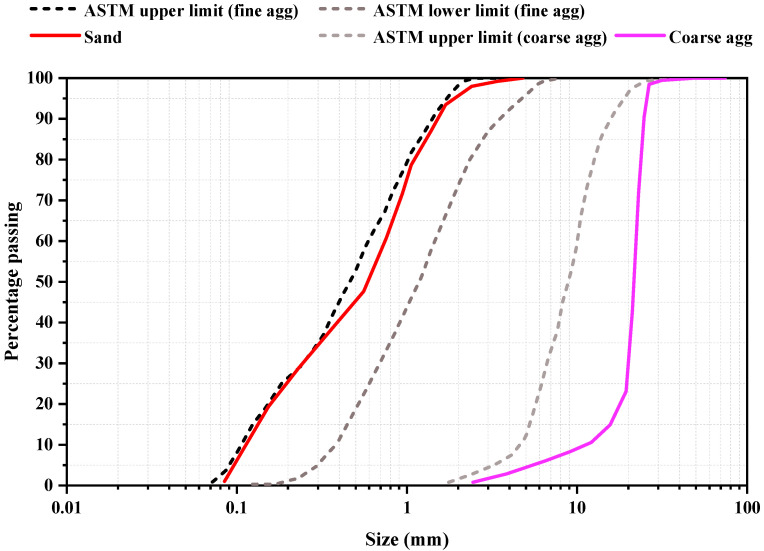
Gradation curve of coarse aggregates and fine aggregates.

**Figure 5 materials-15-08171-f005:**
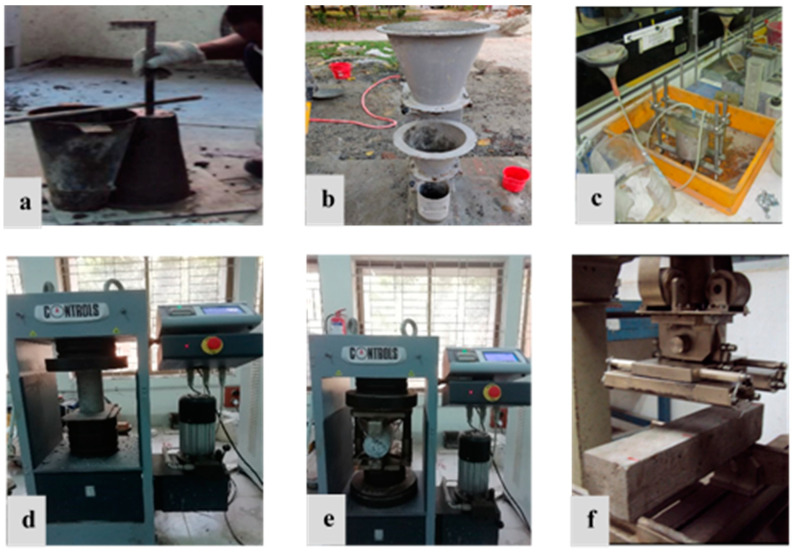
Testing setups: (**a**) slump test, (**b**) compaction test, (**c**) initial surface absorption test, (**d**) compressive strength test, (**e**) splitting tensile strength test, and (**f**) flexural strength test.

**Figure 6 materials-15-08171-f006:**
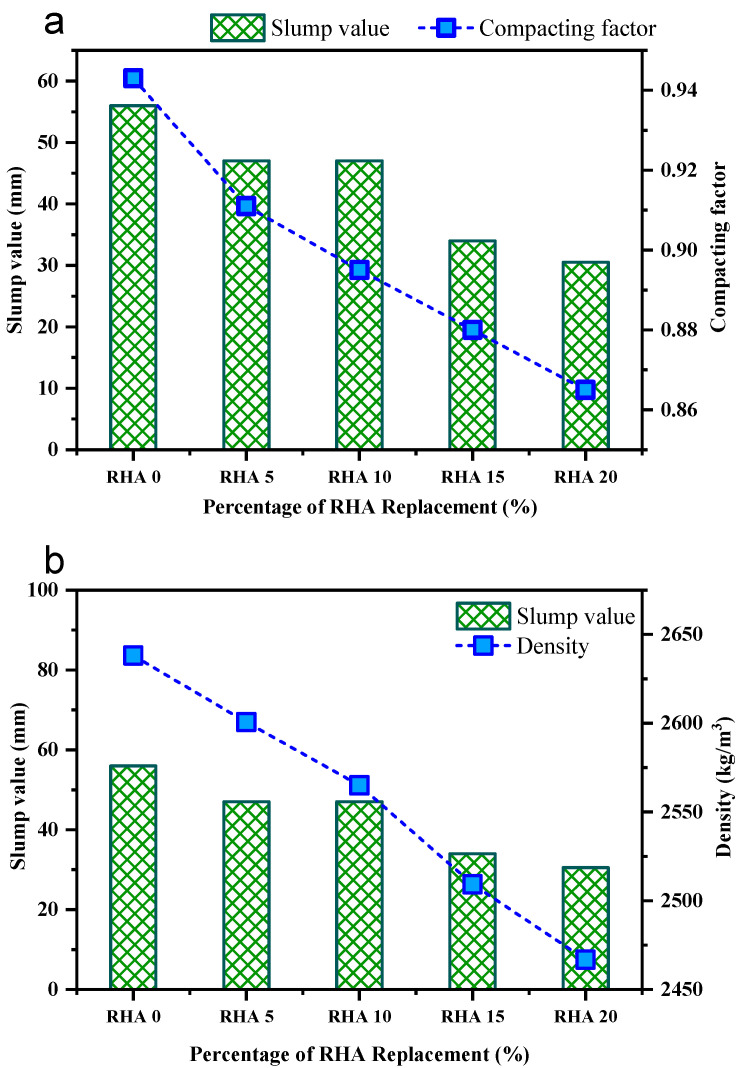
(**a**) Slump values and compacting factor (**b**) variations of slump value and density, with respect to the incorporation of different percentages of RHA.

**Figure 7 materials-15-08171-f007:**
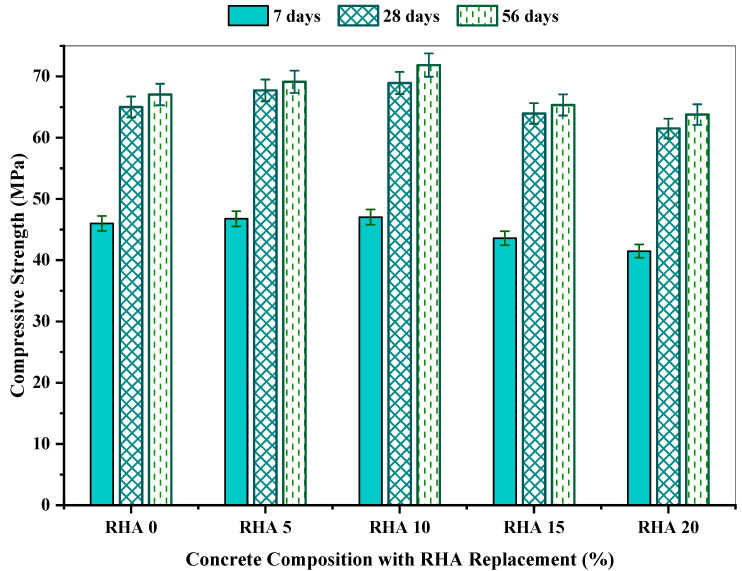
Compressive strength of concrete mixes incorporating RHA at 7, 28, and 56 days.

**Figure 8 materials-15-08171-f008:**
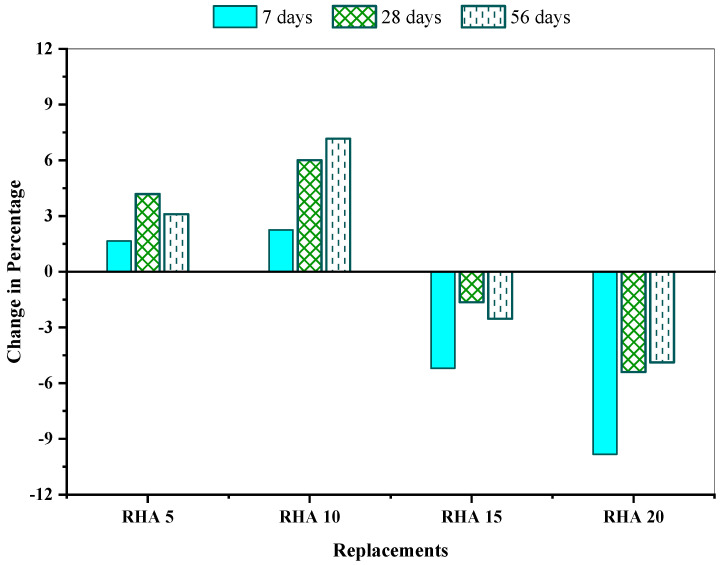
Compressive strength change (%) of the concrete mixes, in comparison to reference concrete.

**Figure 9 materials-15-08171-f009:**
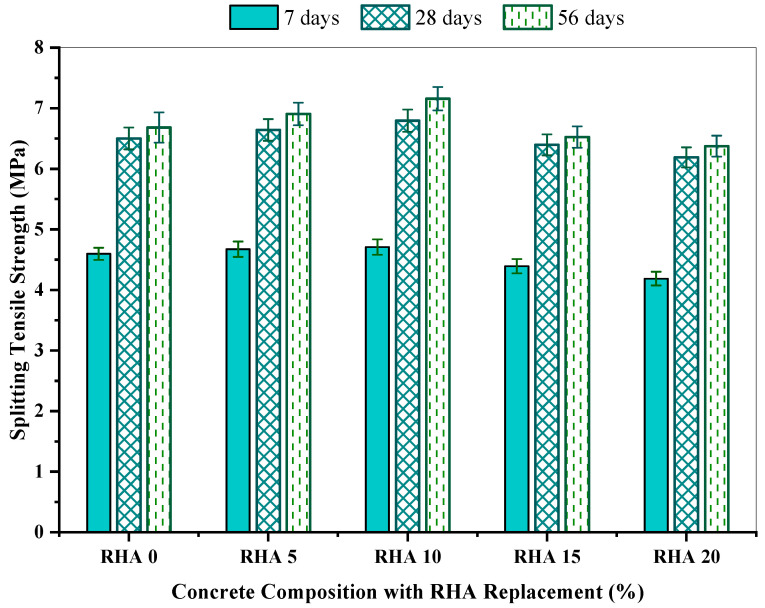
Splitting tensile strength of concrete incorporating RHA at 7, 28, and 56 days.

**Figure 10 materials-15-08171-f010:**
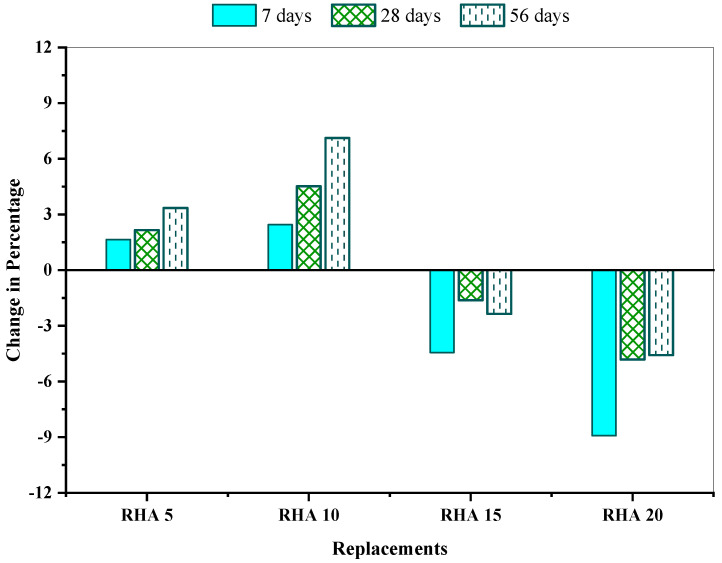
Splitting tensile strength change (%) of the concrete mixes in comparison with reference concrete.

**Figure 11 materials-15-08171-f011:**
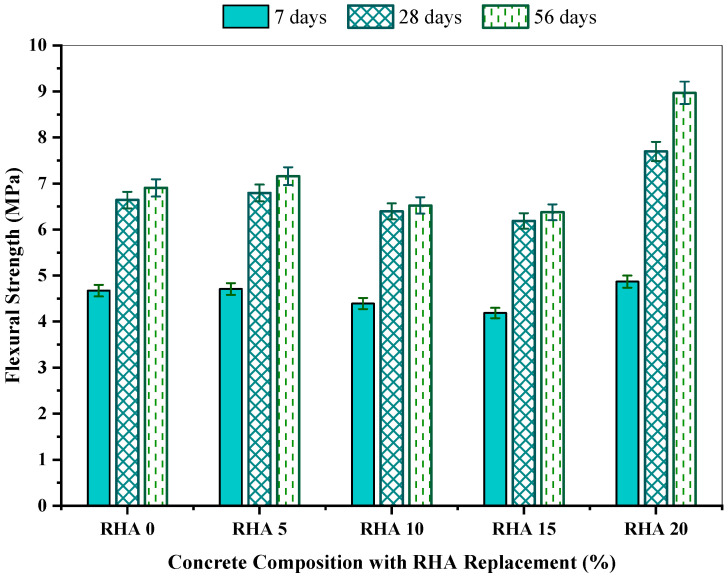
Flexural strength of the concrete incorporating RHA at 7, 28 and 56 days.

**Figure 12 materials-15-08171-f012:**
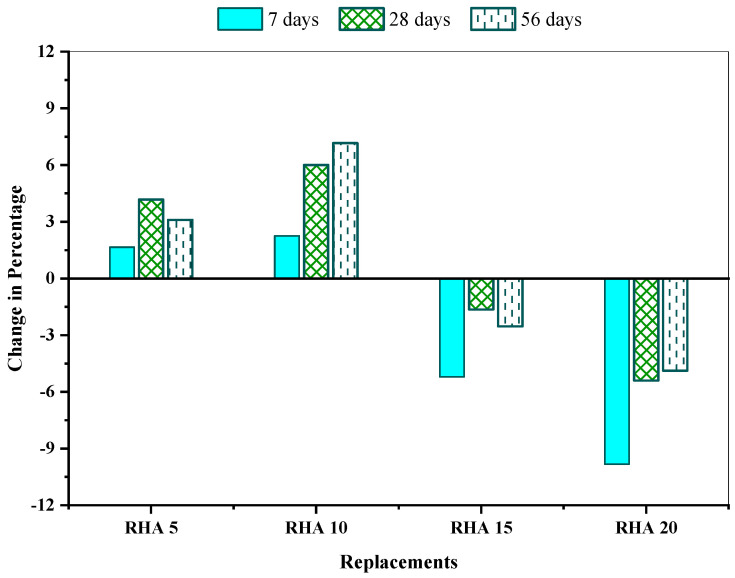
Flexural strength change (%) of the concrete mixes in comparison with the reference concrete.

**Figure 13 materials-15-08171-f013:**
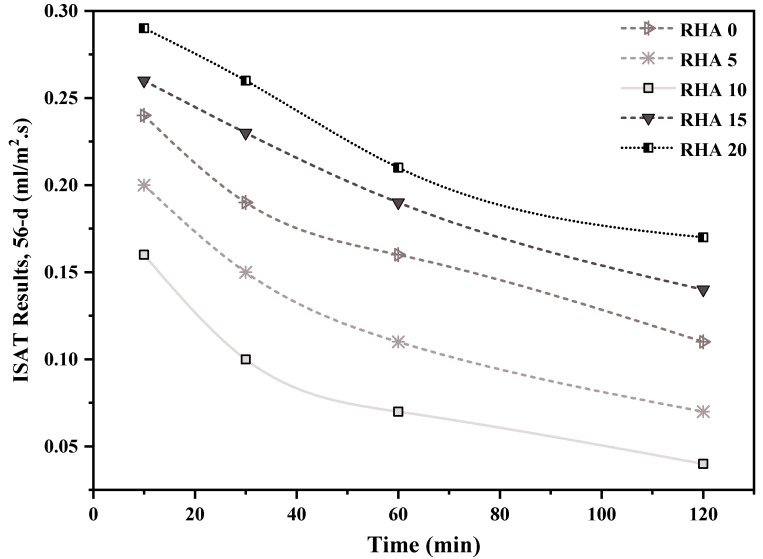
ISAT results, after 56 days, of concrete incorporating different percentages of RHA.

**Figure 14 materials-15-08171-f014:**
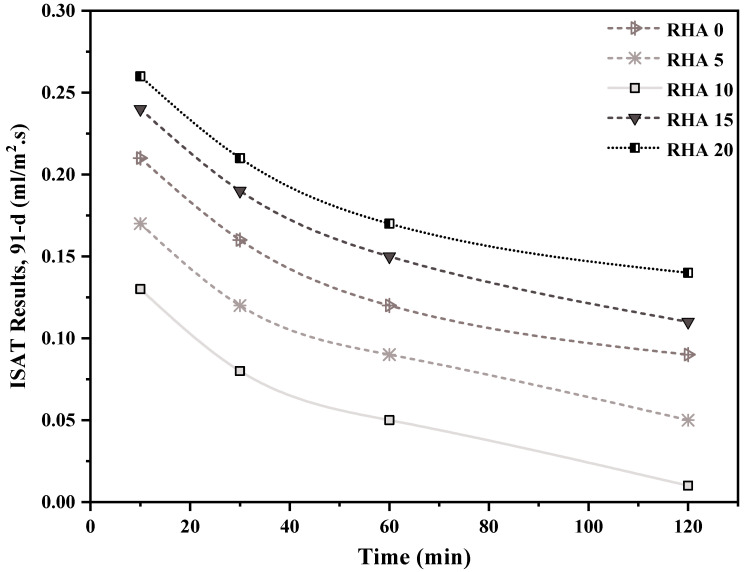
ISAT results, after 91 days, of concrete incorporating different percentages of RHA.

**Figure 15 materials-15-08171-f015:**
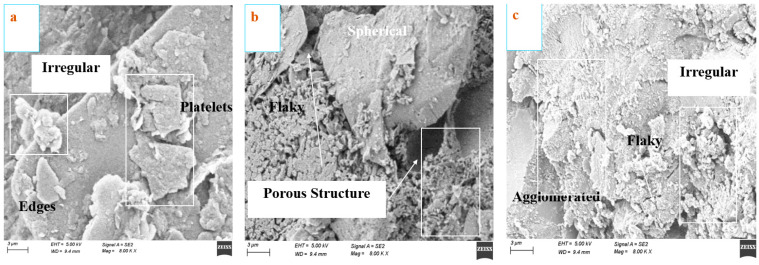
SEM micrographs of concrete with cementitious replacements at 10 µm: (**a**) RHA0 (control), (**b**) RHA10, and (**c**) RHA20.

**Figure 16 materials-15-08171-f016:**
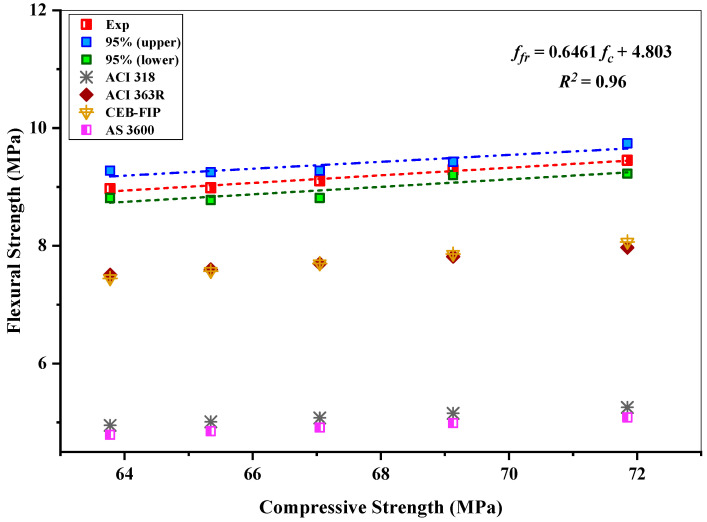
Relationship between the compressive strength and flexural strength of RHA-incorporated concrete mixes.

**Figure 17 materials-15-08171-f017:**
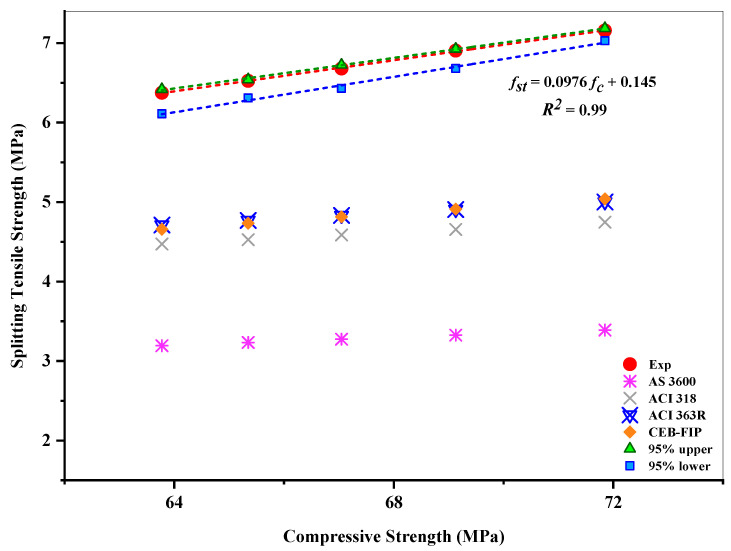
Relationship between the compressive and splitting tensile strengths of RHA-incorporated concrete mixes.

**Figure 18 materials-15-08171-f018:**
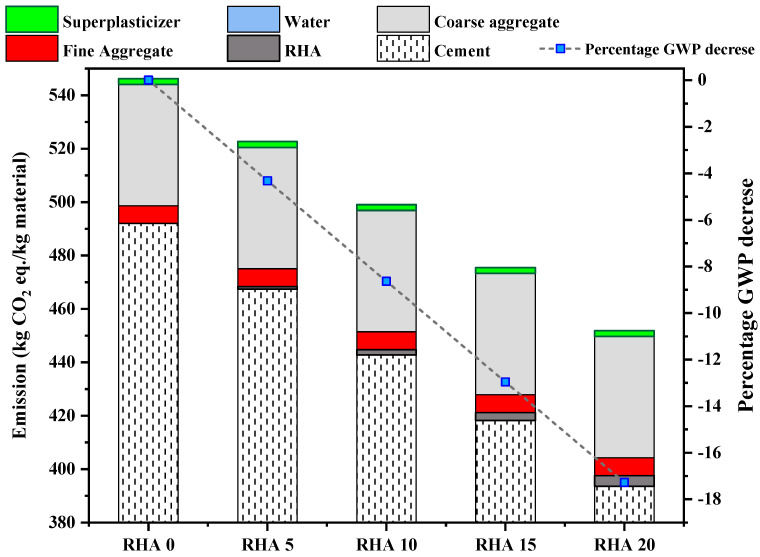
Total CO_2_ emission and GWP enhancement percentage.

**Figure 19 materials-15-08171-f019:**
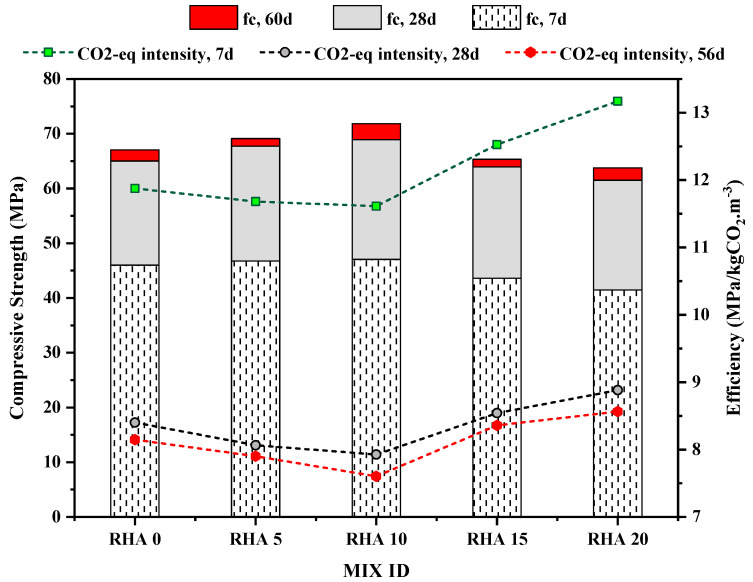
Eco-strength efficiency with respect to compressive strength.

**Figure 20 materials-15-08171-f020:**
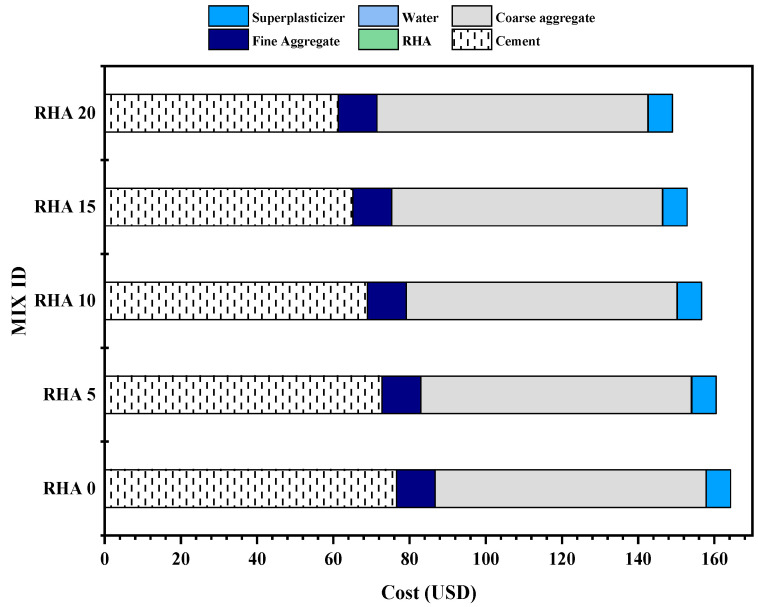
Cost analysis of 1 m^3^ of concrete mixes.

**Figure 21 materials-15-08171-f021:**
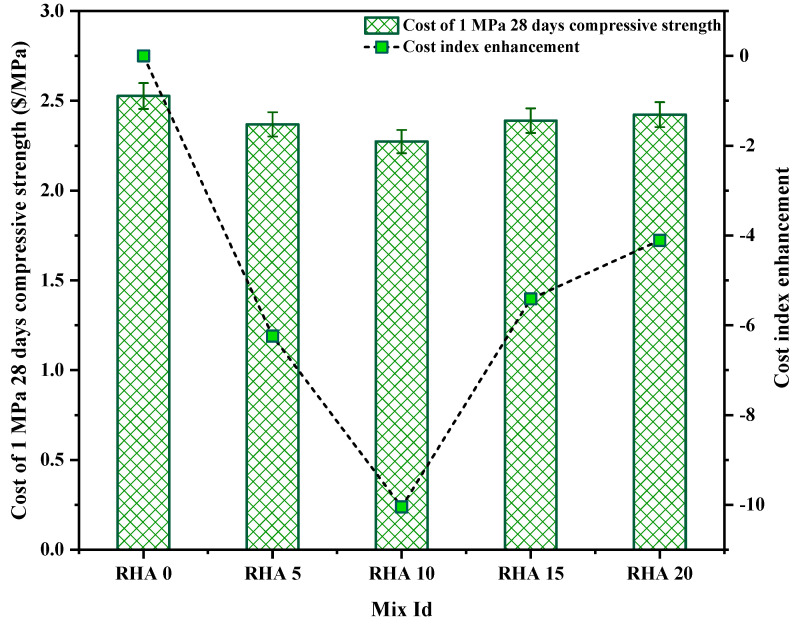
Cost index of the mixes.

**Table 1 materials-15-08171-t001:** Chemical constituents (%) of RHA and OPC.

Oxide	SiO_2_	Al_2_O_3_	Fe_2_O_3_	CaO	MgO	SO_3_	LOI	Na_2_O	K_2_O
OPC	18.8	5.8	4.1	63.5	1.92	2.4	1.24	1.205	2.76
RHA	75.24	2.18	2.24	2.42	2.28	0.12	12.99	0.86	1.72

**Table 2 materials-15-08171-t002:** Physical characteristics of aggregates.

Aggregate Type	Fineness Modulus	Unit Weight (kg/m^3^)	Void (%)	Moisture Content (%)	Water Absorption (%)	Specific Gravity		
						Dry	SSD	Apparent
Coarse	5.94	1684	37.90	1.78	1.14	2.71	2.85	2.88
Fine	2.65	1538	32.23	2.34	1.67	2.45	2.56	2.59

**Table 3 materials-15-08171-t003:** Details of mix proportions of concrete.

Mix ID	Percentage of Replacement (%)	Binder		Fine Aggregate (kg/m^3^)	Coarse Aggregate (kg/m^3^)	Water (kg/m^3^)	SP (kg/m^3^)
		Cement (kg/m^3^)	RHA (kg/m^3^)				
**RHA0**	0% (plain concrete)	600	-	477	1113	165	3
**RHA5**	5%	570	30	477	1113	165	3
**RHA10**	10%	540	60	477	1113	165	3
**RHA15**	15%	510	90	477	1113	165	3
**RHA20**	20%	480	120	477	1113	165	3

**Table 4 materials-15-08171-t004:** Summary of the compressive strength test outcomes.

Mix	Days	Mean Strength (MPa)	Standard Deviation	COV	Standard Error	95% Confidence Interval	
						Lower Range	Upper Range
**RHA0**	7	46.00	0.64	0.01	0.37	45.40	46.66
	28	65.01	0.30	0.00	0.17	64.81	65.36
	56	67.05	0.47	0.01	0.27	66.53	67.46
**RHA5**	7	46.76	0.37	0.01	0.21	46.35	47.05
	28	67.73	0.58	0.01	0.33	67.10	68.24
	56	69.13	0.53	0.01	0.31	68.58	69.63
**RHA10**	7	47.04	0.15	0.00	0.09	46.89	47.20
	28	68.92	0.39	0.01	0.22	68.52	69.29
	56	71.85	0.57	0.01	0.33	71.24	72.37
**RHA15**	7	43.61	0.46	0.01	0.27	43.16	44.09
	28	63.95	0.28	0.00	0.16	63.69	64.24
	56	65.35	0.63	0.01	0.36	64.77	66.02
**RHA20**	7	41.48	0.60	0.01	0.35	40.87	42.08
	28	61.50	0.38	0.01	0.22	61.13	61.88
	56	63.77	1.33	0.02	0.77	62.26	64.74

**Table 5 materials-15-08171-t005:** Summary of the splitting tensile strength test outcomes.

Mix	Days	Mean Strength (MPa)	Standard Deviation	COV	Standard Error	95% Confidence Interval	
						Lower Range	Upper Range
**RHA0**	7	4.60	0.02	0.00	0.01	4.58	4.61
	28	6.50	0.04	0.01	0.02	6.47	6.55
	56	6.68	0.04	0.01	0.03	6.64	6.73
**RHA5**	7	4.67	0.04	0.01	0.03	4.63	4.72
	28	6.64	0.01	0.00	0.01	6.63	6.65
	56	6.90	0.02	0.00	0.01	6.88	6.93
**RHA10**	7	4.71	0.02	0.01	0.01	4.69	4.74
	28	6.79	0.01	0.00	0.01	6.78	6.80
	56	7.16	0.03	0.00	0.02	7.13	7.18
**RHA15**	7	4.39	0.03	0.01	0.02	4.36	4.41
	28	6.40	0.02	0.00	0.01	6.38	6.42
	56	6.52	0.01	0.00	0.01	6.51	6.54
**RHA20**	7	4.19	0.06	0.01	0.03	4.15	4.23
	28	6.19	0.04	0.01	0.03	6.16	6.24
	56	6.37	0.05	0.01	0.03	6.31	6.42

**Table 6 materials-15-08171-t006:** Summary of the flexure strength test outcomes.

Mix	Days	Mean Strength (MPa)	Standard Deviation	COV	Standard Error	95% Confidence Interval	
						Lower Range	Upper Range
**RHA0**	7	5.18	0.17	0.03	0.09	5.05	5.38
	28	7.94	0.32	0.04	0.18	7.73	8.30
	56	9.10	0.25	0.03	0.15	8.81	9.28
**RHA5**	7	5.23	0.01	0.00	0.01	5.23	5.25
	28	8.06	0.19	0.02	0.11	7.85	8.23
	56	9.29	0.12	0.01	0.07	9.20	9.43
**RHA10**	7	5.33	0.04	0.01	0.03	5.28	5.36
	28	8.15	0.09	0.01	0.05	8.10	8.25
	56	9.45	0.26	0.03	0.15	9.23	9.74
**RHA15**	7	4.96	0.28	0.06	0.16	4.70	5.25
	28	7.81	0.29	0.04	0.17	7.55	8.12
	56	8.98	0.24	0.03	0.14	8.78	9.25
**RHA20**	7	4.87	0.08	0.02	0.05	4.78	4.94
	28	7.70	0.06	0.01	0.03	7.64	7.75
	56	8.97	0.27	0.03	0.15	8.81	9.28

**Table 7 materials-15-08171-t007:** CO_2_ emissions data for concrete.

Materials	Emission (kg-CO_2_/kg)	Reference
OPC	0.821	[83]
RHA	0.157	[84]
Sand	0.0139	[85]
Stone Chips	0.0409	[85]
Water	0.000196	[86]
Superplasticizer	0.720	[87]

**Table 8 materials-15-08171-t008:** Local prices of the materials.

Materials	Cement	RHA	Sand	Stone Chips	Water	Superplasticizer
**Cost (USD/kg)**	0.14118	0.00059	0.02353	0.07059	0.00059	2.35294

## Data Availability

Data will be available on appropriate request.

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
