# Peer review of "Integration of Rice Husk Ash as Supplementary Cementitious Material in the Production of Sustainable High-Strength Concrete"

_materials, 2022, doi:10.3390/ma15228171_

Round 1

Reviewer 1 Report

Line 29; Rheology needs to be defined and studied in detail. Authors are discussing fresh properties, not the rheological properties.

During burning of Rice husk, how much energy consumed, and gases released to the environment.  To claim this study as sustainable and environmentally friendly, this needs to be investigated in detail. 

The relation between slump and yield stress needs to be discussed more. Line 204-207. The effect of cone needs to be mentioned. Why normal cone was used in this testing scheme?

A modified cylinder cone model for slump will give better results as compared with normal cone slump. doi.org/10.1016/S0301-7516(02)00148-5 ;  https://doi.org/10.1016/j.cemconres.2003.08.005

Equation 1; what are its limitations, 

GWP calculations need to be mentioned and discussed more. 

Conclusions need to be rewritten and recommendations may be added in it. 

Reviewer 2 Report

Please find the attached word file. 

Reviewer 3 Report

1) In abstract, mention how much strength increase or decrease, give some numbers.

2) Add novelity and originality of this research.

3) Can you put SEM images of used RHA, if possible, to see its microstructure.

4) Can you put more zoom and clear picture of Picture 1.

5) In Figure 7, variation bars are exactly same. Is it.

6) Can you add reasoning for increase or decrease of strength. Also add a separate section for the Discussion, compare your results with previous findings.

7) In the conclusion section, kindly add recommendations based on the conducted study

Reviewer 4 Report

General Comment

The manuscript presents an extensive experimental study to investigate the effect on the properties of both the fresh and hard state of high-strength concrete (HSC) with rice husk ash (RHA) as supplementary material. For this, five concrete mixes were prepared in the laboratory with different percentages of RHA by weight (0%, 5%, 10%, 15% and 20%) and tested to determine the rheological properties at the fresh state (slump, compacting factor and yield stress) and the mechanical properties at the hard state (compressive strength, splitting tensile strength, flexural strength and initial surface absorption). In addition, relationships between strengths are studied and proposed for practice, and a study about the environmental impact and cost benefit is also presented. In the end, the authors conclude about the potential of incorporating RHA from their region to produce a more sustainable HSC.

The manuscripts presents a good literature review on the topic. The raw materials, concrete production and used methodologies are described, including the testing procedures. The results are presented and discussed, including how they compare with the results of previous studies.

The topic of the manuscript is interesting and actual since it is related with the mitigation of energy consumption and pollution related with the production of concrete. The obtained results can highly contribute for this goal and could be very useful for further studies, practitioners and concrete suppliers. The authors have produced a very good research.

I made few comments/suggestions to improve the article. The reviewer encourages the authors to take the suggestions into account and revise their article.

Specific Comment 1

The article should be entirely reviewed to correct some formatting issues. Just one example: some references are not correctly cited in the text. For instance, in page 2 (Section 1) it should be “Ganesan et al. [40]” and not “Ganesan, Rajapogal [40]”. Please revise all the manuscript (please refer to the mdpi instructions for authors).

Specific Comment 2

Section 1, first sentence.

I think that the term “man-made substance” should be substituted by other one in relation with the context of the research. For instance “man-made construction material”.

Specific Comment 3

Section 1

Since several studies already exist on the topic of your research, in the end of Section 1 please clarify better what is the novelty and need of your research.

Specific Comment 4

Section 3.1

In the first sentence, instead of “constructed” it should be better to write “prepared”, as example.

Specific Comment 5

Section 4.1.3

For the sake of the readers, please explain in the manuscript what is the Yield Stress of concrete in the fresh state and what it characterizes.

Specific Comment 6

References

It seems to me that the name of the journal is missing is several references. Please revise and correct the format of the references. Please refer to the mdpi instructions for authors).

Round 2

Reviewer 1 Report

paper is much improved. Reference style need to be checked. some English corrections are required.

Reviewer 2 Report

Thanks the authors for the considerations. The manuscript can be publish now. 

Reviewer 3 Report

Previous comments addressed. References format still need to put as per journal format
